# Irisin and Bone in Sickness and in Health: A Narrative Review of the Literature

**DOI:** 10.3390/jcm11226863

**Published:** 2022-11-21

**Authors:** Elena Tsourdi, Athanasios D. Anastasilakis, Lorenz C. Hofbauer, Martina Rauner, Franziska Lademann

**Affiliations:** 1Department of Medicine III, Technische Universität Dresden Medical Center, 01307 Dresden, Germany; 2Center for Healthy Aging, Technische Universität Dresden Medical Center, 01307 Dresden, Germany; 3Department of Endocrinology, 424 General Military Hospital, 56429 Thessaloniki, Greece

**Keywords:** irisin, FNDC5, osteoporosis, sarcopenia, bone mineral density, fractures

## Abstract

Irisin is a hormone-like myokine produced by the skeletal muscle in response to exercise. Upon its release into the circulation, it is involved in the browning process and thermogenesis, but recent evidence indicates that this myokine could also regulate the functions of osteoblasts, osteoclasts, and osteocytes. Most human studies have reported that serum irisin levels decrease with age and in conditions involving bone diseases, including both primary and secondary osteoporosis. However, it should be emphasized that recent findings have called into question the importance of circulating irisin, as well as the validity and reproducibility of current methods of irisin measurement. In this review, we summarize data pertaining to the role of irisin in the bone homeostasis of healthy children and adults, as well as in the context of primary and secondary osteoporosis. Additional research is required to address methodological issues, and functional studies are required to clarify whether muscle and bone damage *per se* affect circulating levels of irisin or whether the modulation of this myokine is caused by the inherent mechanisms of underlying diseases, such as genetic or inflammatory causes. These investigations would shed further light on the effects of irisin on bone homeostasis and bone disease.

## 1. Introduction and Methodology

Osteoporosis is the most common skeletal disease worldwide, characterized by low bone mass and bone microarchitecture deterioration that could lead to bone fragility and, thus, an increased fracture risk [1,2]. Osteoporosis may occur due to various causes, including menopause or aging (primary osteoporosis), as well as hormonal or metabolic disorders and adverse effects of medications (secondary osteoporosis) [1,2]. At the cellular level, these factors disrupt the balance between bone formation mediated by osteoblasts and bone resorption mediated by osteoclasts leading to impaired bone remodeling and excessive bone degradation. Given that osteoporotic fractures result in severe physical, psychosocial, and financial burdens to both the patient and society, the identification of new targets for drug development is still the main emphasis of basic and translational bone research [3].

In addition to the biomechanical interactions of bone and muscle, recent studies focused on the muscle-to-bone cross-talk mediated by autocrine, paracrine, and endocrine factors [4,5,6]. This close interaction between muscle and bone translates to the clinical level in a second common pathology of the aging population, sarcopenia, i.e., the progressive loss of muscle mass and strength [7]. Common pathophysiological mechanisms, such as physical inactivity, oxidative stress, chronic inflammation, cell senescence, and general changes in body composition, were implicated in the so-called “hazardous duet” of osteoporosis and sarcopenia, often named “osteosarcopenia” [8].

Exercise increases the expression of peroxisome proliferator-activated receptor gamma coactivator 1-alpha (PGC-1a), a transcriptional regulator of fibronectin type III domain-containing protein 5 (FNDC5), in muscle tissue [5]. Irisin is a protein that is cleaved from FNDC5 after skeletal muscle contraction and distributed through the blood (Figure 1). After initially being identified as a critical regulator of adipose tissue browning and thermogenesis, irisin was subsequently found to act on bone [9,10,11,12,13,14,15,16,17,18].

In male mice, the systemic administration of recombinant irisin (r-irisin, 100 µg/kg body weight, weekly, over four weeks) increased cortical bone mass [12], protected against bone loss in a hind-limb unloading model [15], and improved the healing of tibial fractures [13], suggesting osteoanabolic and osteoprotective effects. Furthermore, estrogen and androgen deficiency-induced osteopenia was ameliorated with a systemic r-irisin treatment in rodents [14,19,20]. In line, the osteoprogenitor-specific knockout (Osterix-Cre) of *Fndc5*, the precursor of irisin, reduced trabecular bone mass and delayed skeletal development in mice and revealed bone-derived FNDC5 as a regulator of bone homeostasis [9].

In marked contrast, male *Fndc5* full knockout mice exhibited no bone changes, while female *Fndc5* knockout mice showed trabecular but not cortical bone gain [11]. Challenged by ovariectomy, these *Fndc5*-deficient mice were protected from estrogen deficiency-induced osteoporosis compared to control littermates [11]. Forced *Fndc5* overexpression coupled with a skeletal muscle-specific promotor (muscle creatine kinase, Mck-Cre) led to reduced trabecular bone volume and cortical thickness in young male transgenic mice, identifying muscle-derived FNDC5 as a potential negative regulator of bone mass due to a delayed bone formation rate [18]. In line with *in vivo* studies, *in vitro* experiments also showed controversial outcomes [17,21], implying that the transmembrane receptor FNDC5 itself and its cleavage product, the circulating myokine irisin, might act differentially and independently from each other depending on the targeted cell type and tissue context. Comprehensive reviews on the so-far-known effects of FNDC5/irisin on bone *in vivo* and *in vitro* can be found elsewhere [17,21].

In addition to numerous *in vivo* and *in vitro* studies that have provided evidence of the action of irisin on bone metabolism, extensive clinical data are emerging supporting its physiological relevance on the bone also in humans, although skepticism was expressed with regard to the quality of some of the commercially-available enzyme-linked immunosorbent assay (ELISA) kits used for the measurement of circulating irisin [22]. Significant criticism was raised after the cross-analysis of polyclonal antibodies, which constitute the basis of the available ELISA assays, using Western blotting [23]. These experiments revealed prominent cross-reactivity with non-specific proteins in human and animal sera. Furthermore, an FNDC5 signature was identified with mass spectrometry in human serum but was not detected with the commercial ELISA kits tested in this set of experiments [23]. Conversely, the measurements of the ELISA kits subsequently developed could be validated with Western blotting and mass spectrometry [24,25,26], although the latter methodology has only been implemented in a few human studies to date.

Acknowledging these methodological caveats, in this review, we summarize findings pertaining to the role of irisin in the bone homeostasis of healthy children and adults as well as in the context of primary and secondary osteoporosis. We searched electronic databases (PubMed/MEDLINE) and ClinicalTrials.gov using the MeSH terms “Irisin”, “FNDC5”, “Bone”, and “Osteoporosis”, up to 31 August 2022. As part of the search for this narrative review, we identified 274 abstracts on PubMed. After eliminating duplicates as well as publications that were not pertinent to the subject of irisin and bone status in healthy individuals as well as in subjects suffering from primary or secondary osteoporosis, we retained 37 abstracts, which were included in the final review (Table 1).

## 2. Irisin as a Regulator of Bone Homeostasis in Healthy Children and Adults

Early childhood and adolescence are characterized by rapid bone growth, and the achievement of peak bone mass in young adults affects life-long skeletal health [27,28] and is associated with fracture risk in aging [29]. A few studies have investigated the relationship between irisin and bone homeostasis in childhood. In a population study of 472 pre-pubertal Finnish children, irisin was positively associated with bone mineral density (BMD) independent of body fat mass and lean body mass (LBM) [30]. An Italian study investigated the possible associations of irisin with bone status in healthy children aged 7–13 years and identified a positive correlation of irisin with bone quality as determined with quantitative ultrasonography (QUS) after adjusting for age [31]. In addition, irisin levels positively correlated with serum levels of the bone formation marker osteocalcin and the bone resorption marker cross-linked C-telopeptide of type I collagen (CTX) while being negatively associated with serum concentrations of the Wnt-inhibitor Dickkopf-1 (DKK-1) [31]. These data suggest a putative role of irisin on bone metabolism during childhood and adolescence.

In healthy young adults, circulating irisin displays a day-night rhythm, reaching its nadir concentrations at 6:00 AM and zenith levels at 9:00 PM while being positively correlated with LBM and increasing acutely after exercise [32]. Moreover, different patterns of exercise (short periods of high-intensity exercise vs. moderate continuous exercise) appear to influence irisin secretion [33], with most data suggesting or reporting higher increments of irisin secretion after acute high-intensity exercise [34,35,36]. A few studies have investigated the possible influence of exercise-induced increasing levels of irisin on bone mass. In a study comparing a group of football players to a control group of young individuals with a sedentary lifestyle, footballers were characterized by higher irisin concentrations as well as higher QUS values [37]. Another Italian study of soccer players reported a positive correlation between irisin levels and total body BMD [38]. Of note, this study also identified bone-site specific (i.e., right arm, lumbar vertebrae, and head) correlations between BMD and irisin, thus suggesting an additional systemic effect of irisin on bone [38].

## 3. Irisin in Post-Menopausal and Senile Osteoporosis

Osteoporosis and sarcopenia are two major concurrently occurring clinical entities in the aging population [7,8]. Sarcopenia leads to a progressive loss of muscle mass and strength [7], and muscle wasting could lead to a reduction in myokine production. Indeed, previous studies have identified irisin as a potential biomarker of low muscle mass or sarcopenia [39,40,41]. Analogous to clinical studies investigating the links between irisin and bone homeostasis in young, healthy individuals, research has also targeted irisin as a putative modulator of bone metabolism in the context of age-induced bone disease.

Based on a large cohort of elderly Chinese subjects (N = 6308), Wu and colleagues reported that plasma irisin concentrations were higher in elderly females than males [42]. The authors subsequently implemented an extreme sampling design to identify two subgroups with extremely high (N = 44) and low (N = 36) hip BMDs and reported that plasma irisin concentrations were higher in subjects with high BMDs than subjects with low BMDs, although this association was only seen in males [42]. This sexual dimorphism of plasma irisin with females presenting with higher concentrations had also been reported in previous studies [32,43,44], but the study by Wu et al. highlighted a positive correlation between irisin and BMD in Chinese elderly males [42]. A smaller-scale case-control study of 43 geriatric Chinese men with osteopenia or osteoporosis revealed reduced irisin levels as compared to age-matched controls with normal BMDs and identified irisin as an independent factor affecting BMD via multiple regression analysis [45]. A case-control study of post-menopausal women with osteoporosis and age-matched healthy controls collaborated significantly decreased serum irisin concentrations in women with osteoporosis, although this study did not use multiple regression analysis to identify irisin as an independent regulator of BMD [46]. Of note, a meta-analysis of seven studies with a total of 1018 participants (five studies with post-menopausal women and two studies with both men and women) indicated decreased irisin levels in participants with osteoporosis as well as a positive correlation of irisin with BMD [47].

Further studies have investigated the impact of irisin concentrations on fragility fractures. In a cross-sectional study of 36 overweight subjects having suffered at least one vertebral osteoporotic fracture and 36 overweight controls without osteoporosis, Palermo and colleagues confirmed an inverse correlation between irisin concentrations and vertebral fragility fractures but reported no significant association with BMD [48]. Lower serum irisin levels in post-menopausal women with previous osteoporotic fractures independent of BMD were also found in another study, which reported that a short treatment period with either denosumab or teriparatide did not influence irisin concentrations [49]. Although LBM was not evaluated in this study, the authors postulated that low irisin levels might result from sarcopenic obesity and reduced muscle strength, factors that were associated with osteoporotic fractures [50]. More recent studies have investigated the possible associations of irisin with hip fractures. In a study of 160 elderly women with hip fractures occurring after minimal trauma and 160 aged-matched controls, low irisin concentrations were independently associated with an increased risk of femur fractures when adjusted for BMD or FRAX score [51]. Using a different methodological approach, Liu et al. performed a cross-sectional case-control study in 215 post-menopausal women with hip fractures and 215 age-matched controls without fractures and reported lower serum irisin levels in women with fractures, while identifying irisin levels in the lowest third and fourth quartiles as being positively associated with a high risk of hip fractures and osteoporosis after adjusting for age, BMI, BMD, FRAX, and physical activity score [52]. In contrast, similar irisin levels were reported between women with incident hip fractures sampled right before fixation, and the controls were post-menopausal women with knee or hip osteoarthritis schedules for arthroplasty [53], while even higher irisin levels post-fracture were reported in a study prospectively evaluating 21 patients with low extremity fractures for 60 days [54]. In the latter study, the irisin levels did not change on the first post-operative day compared to its levels before surgery but slightly increased at 15 days and clearly increased at 60 days after surgery [54].

Despite the emerging data of an association between irisin concentrations and osteoporosis and fragility fractures, these observational studies cannot determine a causal relationship. Moreover, circulating irisin levels may not reflect the local actions of the myokine at the bone. To address these limitations, a recent, very elegant study determined the associations of serum irisin with *FNDC5* mRNA in muscle biopsies as well as osteocalcin (*OCN*) mRNA in bone biopsies in osteoporotic and control patients undergoing total hip or knee replacement [55]. This study highlighted that the levels of circulating irisin were consistent with the expression of its precursor *FNDC5* in the skeletal muscle of these subjects, while *FNDC5* in muscle was also positively associated with *OCN* mRNA expression in bone biopsies [55]. In addition, both the serum irisin levels and *FNDC5* expression in the muscle were associated with higher BMDs in this population of elderly adults [55]. Lastly, in a more mechanistic approach, this study showed a significantly increased mRNA expression of the senescence marker *p21* in the bone biopsies of osteoporotic patients, while a recombinant irisin treatment led to a downregulation of *p21* mRNA expression in osteoblasts *in vitro* [55]. Thus, this study suggested a potential senolytic action of irisin, which should be corroborated through the regulation of additional senescence markers through irisin in future human studies.

In conclusion, most human studies have suggested a positive correlation, while others have suggested no correlation between irisin and BMD, while irisin levels were negatively associated with prevalent fractures. In contrast, lower, similar, or even higher irisin levels were reported following hip fractures compared with controls. Most studies have suggested no correlation between irisin and bone turnover markers (Table 1).

## 4. Irisin in Secondary Osteoporosis

### 4.1. Irisin in Diabetic Bone Disease

The pathogenesis of diabetic bone fragility is multi-factorial and appears to be distinct in type 1 diabetes mellitus (T1DM) and type 2 diabetes mellitus (T2DM). Patients with T1DM present with decreased BMDs, possibly because of early-onset bone loss and lower acquired peak bone mass, which poses a risk of osteoporosis and increases the fracture risk across the life span [56,57]. Additionally, poor glycemic control could further aggravate bone homeostasis [58]. In children with T1DM, poor metabolic control resulted in increased serum levels of the Wnt-inhibitors sclerostin and DKK-1 [59]. Observational studies have reported higher plasma irisin concentrations in patients with T1DM compared to euglycaemic controls [60,61]. Regarding the correlation of serum irisin with markers of bone turnover as well as bone density, a study of 96 adolescents with T1DM and 34 controls confirmed increased irisin levels in T1DM patients and confirmed positive correlations of irisin with bone quality as determined with QUS [62]. Of interest, irisin levels correlated negatively with HbA1c in these patients, indicating that irisin could serve as a “predictor” of both bone health and metabolic control in this setting [62].

In contrast to T1DM, patients with T2DM are characterized by normal or increased BMDs compared to age, sex, and BMI-matched controls [63,64], although BMD does not appear to reflect the fracture risk in this population reliably [65]. One of the contributing factors to bone fragility in T2DM is suppressed bone turnover [66]. A small number of studies have indicated decreased irisin concentrations in patients with T2DM compared to euglycaemic controls [67,68], while other studies contested the opposite, namely higher irisin concentrations in the diabetic population [69,70]. Similar findings have emerged in the context of metabolic syndrome, where significantly higher irisin concentrations were reported in comparison to healthy controls [71], giving rise to the theory of irisin resistance in metabolically-compromised individuals [72].

Further studies have investigated the relationship of irisin with BMD and bone turnover markers in T2DM. In a case-control study of 66 new-onset T2DM patients and 82 control subjects, the circulating levels of bone turnover markers and irisin were lower in T2DM patients [73]. Moreover, a sub-group analysis revealed reduced irisin levels among T2DM patients with osteoporosis compared to diabetics with normal BMDs, suggesting that irisin could be a potential biomarker of diabetic bone disease in the initial stages of the diagnosis of the disease [73]. In contrast, a cross-sectional case-control study of 83 patients with T2DM and 81 euglycaemic controls revealed significantly higher irisin concentrations in diabetic patients, while a weak association of irisin with physical activity was noted in healthy subjects but not in diabetics [74]. It is possible that metabolic abnormalities known to influence the exercise-dependent expression of PGC-1a [75,76] could also translate into the impairment of PGC-1a-dependent *FNDC5* expression. Lastly, in the setting of pre-diabetes as defined by either impaired fasting glucose or impaired glucose tolerance, serum irisin was shown to be inversely associated with serum sclerostin, independent of age, gender, and BMI, thus highlighting another facet in the cross-talk of muscle and bone in metabolically impaired subjects [77].

### 4.2. Irisin in Primary Hyperparathyroidism

Primary hyperparathyroidism (PHPT) is an endocrine disorder characterized by hypercalcemia due to the aberrant production of parathyroid hormone (PTH), and although most often, the presentation of PHPT is asymptomatic in the regions of the world where serum levels of calcium are routinely measured, more severe cases PHPT can manifest osteoporosis and hypercalciuria as well as vertebral fractures and nephrolithiasis [78]. In the skeleton, PHPT is characterized by a predominant loss of BMD at cortical sites, such as the distal one-third of the forearm, whereas the lumbar spine, which mostly comprises trabecular bone, is relatively spared [79]. This pattern reflects the catabolic versus anabolic effects of PTH on different skeletal sites [80]. Indeed, *in vitro* studies have revealed that chronically elevated PTH concentrations stimulate osteoclastogenesis indirectly via the release of nuclear factor receptor-κB ligand (RANKL) through osteoblasts [81]. On the other hand, intermittent PTH treatments enhance bone formation by targeting the osteoblast through transcriptional changes in several pathways [81].

Recent preclinical and clinical studies have suggested possible interactions between irisin and PTH [82,83]. At a cellular level, the treatment of C2C12 myotubes with terirapatide downregulated the expression of *Fndc5*, while the exposure to recombinant irisin reduced PTH receptor mRNA expression in MC3T3-E1 osteoblasts, highlighting the opposite actions between the two hormones in muscle and bone cells *in vitro* [84]. This study suggested that irisin could exert its osteoanabolic effect not only with the stimulation of osteoblast formation but also by the mitigation of the catabolic action of PTH in these cells [84]. Further highlighting the interplay between PTH and irisin, post-menopausal women with PHPT displayed lower irisin serum concentration compared to aged-matched healthy controls [84]. These findings are concordant with those of a previous clinical study showing a negative correlation of irisin with PTH in post-menopausal women with low bone mass [49].

### 4.3. Irisin in Cushing’s Disease

Cushing’s disease (CD), the most prevalent cause of endogenous hypercortisolism, is the consequence of prolonged exposure to high levels of cortisol because of an ACTH-secreting tumor of the pituitary [85]. CD is characterized by a wide array of clinical manifestations, including metabolic, cardiovascular, psychiatric, and musculoskeletal complications [86]. One of the most common clinical findings in patients with CD is muscle atrophy and weakness predominantly affecting the proximal muscles of the lower extremities [87]. Patients suffering from CD develop sarcopenia concurrently with obesity and osteoporosis, also known as osteosarcopenic obesity. Given the multi-faceted actions of irisin on the skeletal muscle, bone, adipose tissue, and glucose homeostasis, which are also the target organs of cortisol, it appears plausible that excess cortisol could result in changes in irisin levels. A recent study assessed the circulating irisin concentrations in patients with CD during the active phase of the disease and after biochemical remissions compared to controls [88]. Irisin levels were lower in patients with active CD when compared to both healthy controls and patients in remission. Multiple linear regression analysis revealed strong associations of irisin with metrics of osteosarcopenia and central obesity in CD [88]. Interestingly, this study also confirmed a reverse correlation between irisin and PTH, thus supporting the data reported above [49,84]. Further investigations should assess whether irisin could act as a metabolic biomarker of osteosarcopenic obesity in patients with CD.

### 4.4. Irisin in Growth Hormone Deficiency

Growth hormone deficiency (GHD) is caused by a lack of pituitary-derived growth hormone (GH) and, when it presents in childhood, causes an abnormally short stature with normal bone proportions and low BMD [89]. Adults with GHD display abnormalities in their body composition, i.e., increased obesity, impaired skeletal muscle performance, and a decreased BMD accompanied by a higher risk for fragility fractures [90], and GH replacement could counteract these manifestations [91]. Since irisin is involved in adipose tissue homeostasis and glucose metabolism, changes in irisin could mediate the effects of GHD. Pre-pubertal children with idiopathic GHD were characterized by a lower growth velocity, higher body mass index (BMI), and significantly lower serum irisin concentrations when compared to healthy short children [92]. In the same cohort, GH replacement therapy of GH-deficient children over 12 months led to an increase in irisin levels and decreased BMI [92], although the irisin concentrations did not reach the values of the control subjects.

### 4.5. Irisin in Hypothalamic Amenorrhea

Functional hypothalamic amenorrhea (FHA) is defined as the absence or cessation of menstrual cycles as the result of the suppression of the hypothalamus-pituitary-ovary axis without concurrent organic injury. Common causes of FHA include stress, marked weight loss, very low BMI, and excessive exercise [93]. Although potentially reversible, FHA is characterized by hypogonadism that could lead to a progressive risk of developing osteopenia or osteoporosis [94]. The co-existence of low energy availability (with or without eating disorders), amenorrhea or oligomenorrhea, and a reduced BMD is known as “female athlete syndrome” [95]. Given the well-established action of irisin as an inducer of white adipose tissue browning [96] and the positive association between brown adipose tissue and total bone area [97], it seems plausible to expect changes in irisin in subjects with FHA. A study in young amenorrheic athletes with FHA confirmed lower irisin concentrations in comparison to athletes without amenorrhea and non-athletes, suggesting that a decrease in irisin could represent an adaptive response to conserve energy in these individuals [98]. Irisin levels were positively associated with metrics of BMD and bone strength and structure in both athletes across all groups [96]. A more recent study revisited the subject of irisin changes in FHA, removing the factor of physical exercise [99]. Patients with FHA displayed significantly lower BMIs and irisin concentrations when compared to healthy aged-matched controls associated with lower BMDs, which, however, did not reach the osteopenic range [99]. Of note, a similar decrease in irisin levels has not been reported in the context of anorexia nervosa (AN), which appears counterintuitive. In a study of 42 young women with restrictive AN, the serum irisin concentrations did not differ from those of age-matched controls, nor were these associated with resting energy expenditures [100]. On the other hand, these young women presented with expected lower BMD values, as well as an up-regulation of bone resorption and a decrease in bone formation [100]. The differences in energy expenditure rates between amenorrheic athletes and patients with AN could explain the distinct results in irisin levels, although this should be further investigated.

### 4.6. Irisin in Chronic Kidney Disease

Osteoporosis and fractures are common in people with advanced chronic kidney disease (CKD) and on maintenance dialysis and are associated with high all-cause mortality [101]. Chronic kidney disease-mineral bone disorders (CKD-MBDs) promote not only bone disease (osteoporosis and renal dystrophy) but also vascular calcification and cardiovascular disease. In the setting of CKDs, irisin deficiency was shown to contribute to the development of endothelial dysfunction and vascular calcification [102], and a number of studies have established a decrease in irisin concentrations in patients with advanced CKDs [39,103,104]. In patients on maintenance hemodialysis, their serum irisin levels were positively correlated with lumbar spine BMD independently of their baseline comorbidities and dialysis duration, and patients with end-stage renal disease and osteopenia or osteoporosis displayed lower irisin levels than patients with normal BMD [105]. A very recent study including patients with hemodialysis or peritoneal dialysis confirmed the decreased irisin concentrations in this cohort compared to healthy controls, while the bone mineral content, as assessed with segmental bioelectric impedance, revealed no significant difference in the bone mineral content between the two groups [106]. Significant inverse relationships were found between irisin and alkaline phosphatase as well as between irisin and PTH in the patient group, suggesting that irisin simultaneously inhibits bone resorption and protects against vascular calcification [106].

### 4.7. Irisin in Rheumatoid Arthritis

Rheumatoid arthritis (RA) is a chronic inflammatory autoimmune disease that primarily affects synovial joints leading to joint destruction, loss of function, and disability [107]. A number of inflammatory cytokines, most notably interleukin-6 (IL)-6, interleukin-1β (IL-1β), and tumor necrosis factor-α (TNF-α), were identified as major players in the pathogenesis of RA as well as RA-related comorbidities, such as osteoporosis [108]. The prevalence of osteoporosis in RA patients was documented to vary from 19% to 30% [109,110], with a prevalence of osteoporotic vertebral fractures in RA patients ranging from 13% to 40% [111,112]. Exercise was identified as an important non-pharmacological intervention in chronic inflammatory diseases such as RA by modifying the inflammatory cytokine milieu [113]. As physical exercise is a well-known stimulant of irisin [34,35,36], a number of recent studies have investigated the role of irisin in RA-induced bone disease. In RA patients, serum irisin concentrations were significantly decreased and associated with the presence of fragility fractures, higher grades of inflammatory activity, longer disease durations, and the presence of extra-articular manifestations [114]. Similar results were reported in a more recent study, where patients with RA and vertebral fractures displayed lower serum irisin concentrations compared to RA patients who did not have fractures [115]. Using logistic regression analysis, this study revealed that the increased risk of vertebral fractures related to low irisin levels was independent of other factors, including age, low BMI, a longer disease duration of RA, and disease activity [115]. Using a different approach, Ercan et al. aimed to investigate the effects of acute aerobic exercise on inflammatory markers and irisin in patients with RA compared to controls [116]. Counterintuitively, patients with RA depicted higher irisin levels than healthy controls at the baseline, which was suggested to be a compensatory mechanism to counteract the inflammatory milieu of RA, while the irisin levels decreased in both groups post-exercise [116]. Further studies are needed to prove whether irisin could constitute a reliable biomarker in RA-induced bone disease.

### 4.8. Irisin in Genetic Syndromes

#### 4.8.1. Turner Syndrome

Turner syndrome (TS) is a chromosome disorder characterized by a female phenotype caused by either complete or partial loss of one X chromosome, often in mosaic karyotypes. The manifestations of TS include a short stature, delayed puberty, gonadal dysgenesis, hypergonadotropic hypogonadism, infertility, congenital malformations of the heart, endocrine disorders, such as T1DM and T2DM, osteoporosis, and autoimmune disorders [117]. Of note, patients with TS often display abdominal obesity, and the basic metabolic defect in TS was suggested to be insulin resistance [118]. Furthermore, patients with TS are resistant to the physiological levels of GH, so they often require supraphysiological doses of GH replacement therapy [117]. Given the multi-faceted metabolic complications and alterations of body compositions and linear growth in TS, it is of interest to investigate the possible changes of irisin in this disease. In a study of 36 girls with TS treated with recombinant GH, this treatment not only improved their final height and body composition but also led to a significant increase in plasma irisin concentrations [119]. Although the pathogenesis of growth impairment in TS is not thoroughly elucidated, it was suggested that the partial insensitivity of bone fibroblasts to the action of GH could play a major role [120], and it remains to be investigated whether irisin is also implicated in the short stature of patients with TS.

#### 4.8.2. Prader-Willi Syndrome

Prader-Willi syndrome (PWS) is a rare genetic disease associated with childhood-onset obesity, hyperphagia, GHD, hypogonadism, dysmorphic features, cognitive impairments, and behavioral problems [121]. PWS is also characterized by bone impairment due to the failure of peak bone mass attainment secondary to GHD and hypogonadism [122]. Consequently, adults with PWS could present with osteoporosis and fractures while also having other orthopedic complications of excess weight [123]. Salivary irisin concentrations were increased in obese PWS patients compared to non-obese controls, although this finding was not reflected in the serum irisin concentrations of this cohort [124]. In a recent study of 78 patients with PWS (26 children and 52 adults), the irisin serum concentrations did not differ from those of healthy controls [125]. Of note, this study revealed that patients not receiving vitamin D supplementation had lower irisin levels [125]. Via multiple regression analysis, irisin levels could be predicted by BMD, the genetic background and 25(OH) vitamin D levels in pediatric and adult PWS [125].

#### 4.8.3. Charcot-Marie-Tooth Disease

Charcot-Marie-Tooth disease (CMT) comprises a group of rare hereditary disorders that are characterized by the progressive neuropathy of the motor and sensory nerves [126]. The clinical manifestations include slowly progressive distal weakness, muscle atrophy, loss of peripheral reflexes, and skeletal deformities [126]. Currently, treatments are restricted to symptomatic pain relief [127], while the rehabilitative management of patients with CMT could provide moderate improvements in muscle strength and function [128]. A recent study assessed the irisin levels in 20 CMT patients and reported significantly lower serum irisin concentrations in this cohort compared to healthy controls matched for age, sex, and BMI [129]. These patients displayed below-normal muscle qualities, as evaluated with the muscle strength and muscle-mass ratios, and in a regression analysis model, irisin was a positive determinant of the bone formation marker P1NP, suggesting a role of this myokine in the regulation of bone mass in CMT [129].

#### 4.8.4. Becker Muscular Dystrophy

Becker muscular dystrophy is an X-linked inherited genetic disease caused by a pathogenic variant coding for the dystrophin gene and resulting in lower but detectable dystrophin expression in muscle fibers [130]. Although both Duchenne and Becker muscular dystrophy are caused by mutations in dystrophin, in Duchenne muscular dystrophy, functioning dystrophin is completely absent in the muscle, while in Becker muscular dystrophy, there are lower amounts of dystrophin present, although not enough for completely normal muscle function [130]. Patients with Duchenne muscular dystrophy gradually develop severe bone loss secondary to muscle impairment following immobilization [131]. In addition, these patients display altered whole-body composition, with a decrease in lean bone mass correlating with a loss of muscle function and strength [132]. In patients with myotonic dystrophy, their irisin levels were significantly lower than in healthy controls [133], suggesting a putative role of irisin as a potential marker for muscle dysfunction [134]. A recent study confirmed significantly lower irisin levels in patients with Becker muscular dystrophy compared to healthy controls, while irisin was negatively correlated with lower extremity BMD in this cohort, indicating the resistance to irisin action at the bone tissue in this setting [135].

**Table 1 jcm-11-06863-t001:** Human studies evaluating associations of irisin with bone parameters.

Name, Year [Ref.]	Study Type	Population	Main Outcomes				Comments
			BMD (DXA)	BTM	Fractures	Other	
Soininen, 2018 [30]	cross-sectional baseline analysis of cohort study	472 pre-pubertal children, 6–8 yrs (245 M-227 F)	(+) correlation of irisin with WB BMD	NM	NR		association was lost separately for F and M
Colaianni, 2019 [31]	cross-sectional	34 children, 10 ± 3 yrs	NM	(+) correlation of irisin with OC and CTX (−) correlation with Dkk1	NR	(+) correlation of irisin with Ad-SoS and BTT Z-score (QUS)	no DXA data
Gaudio, 2021 [37]	cross-sectional	15 M footballers vs. 15 M sedentary controls	NM	NM	NR	higher irisin, RANKL, and stiffness (QUS) in footballers	-no DXA data -no correlations between irisin and QUS parameters or BTM were reported
Colaianni, 2017 [38]	cross-sectional	18 M footballers, 24.7 ± 1.2 yrs	(+) correlation of irisin with WB BMD and with specific bone-site BMD (right arm, lumbar vertebrae, and head)	NM	NR		-no control group
Wu, 2018 [42]	cross-sectional analysis of archived cohort study	80 elderly Chinese (44 with extremely high vs. 36 with extremely low TH BMD)	(+) correlation of irisin with TH BMD in M but not F	no correlation of irisin with P1NP or CTX	NR		
Zhang, 2020 [45]	case-control	67 elderly Chinese M (43 osteopenia/OP vs. 24 normal BMD	-lower irisin in osteopenia/OP vs. normal BMD -(+) association of irisin with BMD	(+) correlation of irisin with P1NP	NR		-not clarified if the correlation was with LS, FN, BMD, or both
Engin-Ustun, 2016 [46]	cross-sectional	172 PM F (88 osteoporosis vs. 88 normal BMD)	-lower irisin in OP vs. normal BMD	NM	NR		-no BMD data reported -no correlations between irisin and BMD were reported
Palermo, 2015 [48]	cross-sectional	72 PM F, 64.3 ± 6.1 yrs (36 OP + VFx vs. 36 osteopenia or normal BMD + no VFx)	- no correlation of irisin with LS, FN, or TH BMD	NM	lower irisin in patients (OP + VFx)		-all patients (osteoporosis + VFx) were treated with Dmab or TPTD
Anastasilakis, 2014 [49]	case-control	125 PM F (75 with T-score ≤ −2.0 vs. 50 with T-score > −2.0)	-no correlation of irisin with LS or FN BMD	-no correlation of irisin with CTX	-lower irisin in Fx regardless of BMD -(−) association of irisin with prevalent Fx	-irisin (−) associated with PTH -irisin was not affected by short-term TPTD or Dmab treatment	
Yan, 2018 [51]	cross-sectional, case-control	320 Chinese F, 72–80 yrs (160 incident hip Fx vs. 160 matched controls without Fx)	(+) correlation of irisin with LS and FN BMD	-no correlation of irisin with CTX or P1NP	-lower irisin and (−) association with hip Fx after adjustment for BMD or FRAX		
Liu, 2021 [52]	cross-sectional, case-control	430 PM Chinese F, mean 68.7 yrs (215 incident hip Fx vs. 215 matched controls without Fx)	(+) correlation of irisin with WB and TH BMD	-no correlation of irisin with CTX or P1NP	-lower irisin in hip Fx and (−) association with hip Fx	(−) association of irisin with osteoporosis risk	no correlation of irisin with LS or FN BMD
Anastasilakis, 2021 [53]	cross-sectional, case-control	69 PM F (32 incident hip Fx vs. 37 knee or hip OA)	-no correlation of irisin with LS or FN BMD	-no correlation of irisin with CTX or P1NP	-no association of irisin with hip Fx		
Serbest, 2017 [54]	cohort study	21 adult patients with hip or tibia Fx	NM	NM	NR	compared to before surgery, irisin was unchanged on the 1^st^ day, slightly increased at 15 days, and clearly increased at 60 days post-operative	-no control group
Colaianni, 2021 [55]	cross-sectional	62 patients, 68.7 ± 12.3 yrs (16M-46F) undergoing hip or knee arthroplasty	-(+) correlation of irisin with LS and TH BMD -irisin lower in osteopenia/OP vs. normal BMD	-no correlation of irisin with CTX	NR	irisin levels (+) associated with FNDC5 expression in muscle and FNDC5 expression in muscle (+) correlated with OC mRNA in bone - recombinant irisin downregulated the senescence marker *p21* mRNA expression in osteoblasts	
Faienza, 2018 [62]	cross-sectional	96 T1DM children/adolescents, 12.2 ± 4.0 yrs (41M-55F) vs. 34 controls, 9.8 ± 3.4 yrs (21M-13F)	NM	-(+) correlation of irisin with CTX and OC	NR	-(+) correlation of irisin with BTT Z-score (QUS) - irisin (+) correlated with PTH - higher irisin in T1BM but (−) correlated with HbA1c	-no DXA data
Wang, 2022 [73]	cross-sectional, case-control	66 Chinese new-onset T2DM, 51.1 ± 12.7 yrs vs. 82 controls, 50.0 ± 9.0 yrs	-no correlation of irisin with LS BMD	-(−) association of irisin with CTX -no correlation of irisin with P1NP or OC	NR	-lower irisin in T2DM -lower irisin in T2DM with OP vs. T2DM with normal BMD	
Palermo, 2019 [84]	cross-sectional	26 PM F with PHPT, 65.8 ± 7.9 yrs vs. 31 controls, 64.2 ± 5.8 yrs	-no association of irisin with LS, FN, TH, or radial BMD	NR	NR	-lower irisin in PHPT	-no correlations of irisin with BTM were reported
Singhal, 2014 [98]		85 adolescent F [38 athletes with FHA(AA) vs. 24 eumenorrheic athletes(EA) vs. 23 non-athletes(NA)]	-(+) association of irisin with LS, FN, and WB BMD Z-scores	-no association with CTX or P1NP		-(+) correlation of irisin with vBMD and bone strength (HRpQCT and FEA) -lower irisin in AA vs. EA and NA	
Notaristefano, 2022 [99]	retrospective observational	32 FHA, 18–34 yrs, vs. 19 matched controls	-(+) association of irisin with TH BMD but (−) with LS and FN BMD	NM	NR	-lower irisin in FHA	
Maimoun, 2022 [100]	cross-sectional, case-control	42 F with RAN vs. 42 controls, 18.5 ± 4.2 yrs	-no correlation of irisin with LS, FN, TH, or radius BMD	-no correlation with CTX or P1NP	NR	-irisin similar in RAN and controls	
Lu, 2021 [105]	cross-sectional	80 HD patients, 66.9 ± 10.3 yrs, 41M-39F (51 normal BMD vs. 19 osteopenia vs. 10 OP)	-(+) association of irisin with LS BMD	-(+) association of irisin with TALP	NR		
Csiky, 2022 [106]	cross-sectional	52 HD, 56.5 ± 11.9 yrs (24M-28F) and 15 PD, 55.8 ± 14.7 yrs (8M-7F) vs. 37 controls, 54.8 ± 4.7 yrs (8M-29F)	NM	-(−) correlation of irisin with TALP	NR	-irisin decreased in HD/PD vs. controls -no correlation of irisin with BMC(BIA)	
Lavrova, 2018 [114]	cross-sectional	110 RA vs. 60 controls	NM	NM	-relation of irisin with low-energy fractures		-only abstract/full article could not be retrieved
Gamez-Nava, 2022 [115]	cross-sectional	148 RA F, 59.0 ± 10.0 yrs vs. 97 control F	NR	NM	-lower irisin in RA with VFx vs. RA without VFx -(+) association of irisin with VFx risk		-no correlations of irisin with BMD were reported
Faienza, 2021 [125]	cross-sectional, case-control	78 PWS patients, [26 children, 9.48 ± 3.6 yrs (11M-15 F) and 52 adults, 30.6 ± 10.7 yrs (22M-30F)] vs. 80 controls (26 children and 54 adults)	-(+) association of irisin with LS BMD Z-score (children) and T-score (adults)	NR	NR	-similar irisin in PWS vs. controls -irisin was negatively associated with PTH	
Colaianni, 2022 [129]	cross-sectional	20 CMT, 54.0 ± 14.5 yrs vs. 20 historical controls	NM	-(+) correlation of irisin with P1NP/no correlation with CTX or OC	NR	-(−) association of irisin with 25OHD -no correlation with sclerostin, OPG, or RANKL -lower irisin in CMT vs. controls	
Barp, 2022 [135]	cross-sectional	29 Becker muscular dystrophy patients vs. 20 healthy controls	-(−) correlation of irisin with lower limbs BMD	NR	NR	-lower irisin in Becker muscular dystrophy vs. controls	-no correlations of irisin with BTM were reported

**Abbreviations:** (+), positive; (−), negative; 25OHD, 25-hydroxyvitamin D; Ad-SoS, amplitude-dependent speed of sound; BALP, bone-specific alkaline phosphatase; BIA, bioelectric impedance; BMC, bone mineral content; BMD, bone mineral density; BTM, bone turnover markers, BTT, bone transmission time; CMT, Charcot-Marie-Tooth Disease; CTX, C-terminal telopeptide of type 1 collagen; Dkk1, dickkopf-1; DXA, dual energy X-ray absorptiometry; F, female; FEA, finite element analysis; FHA, functional hypothalamic amenorrhea; FN, femoral neck; Fx, fracture; HD, hemodialysis; HRpQCT, high resolution peripheral quantitative computed tomography; LBM, lean body mass; LS, lumbar spine; M, male; non-VFx, non-vertebral fracture; NM, not measured; NR, not reported; OA, osteoarthritis; OC, osteocalcin; OP, osteoporosis; P1NP, N-terminal propeptide of type 1 collagen; PHPT, primary hyperparathyroidism; PM, post-menopausal; PD, peritoneal dialysis; PWS, Prader-Willi syndrome; QUS, quantitative ultrasound; RA, rheumatoid arthritis; RAN, restrictive anorexia nervosa; T1DM, type 1 diabetes mellitus; T2DM, type 2 diabetes mellitus; TALP, total alkaline phosphatase; TH, total hip; vBMD, volumetric BMD; VFx, vertebral fracture; WB, whole body; yrs, years.

## 5. Conclusions

Irisin is a myokine with multi-faceted actions in energy metabolism in physiological and pathological conditions [136], and emerging data have highlighted its role on the muscle-bone axis in particular (Figure 1). Recent evidence suggests that irisin acts on target cells, including bone cells [11]. Emerging data indicate that irisin could be part of the numerous autocrine and paracrine factors that regulate the activity of osteoblasts, osteoclasts, and osteocytes. However, fundamental questions about the cleavage mechanisms of irisin and whether FNDC5 might act differentially and independently from irisin on bone metabolism remain unanswered.

In humans, most studies have shown reduced serum irisin levels with aging and in the context of bone disease, both in the setting of primary and secondary osteoporosis. However, it should be noted that recent reports have questioned the significance of circulating irisin, claiming that the detection of this myokine via ELISA, Western blotting, and mass spectrometry is compromised by methodological issues and does not provide reliable and reproducible data [22,24]. Thus, the role of irisin in humans is still a matter of debate [137]. Further methodological studies are needed to address these technical issues, and functional studies are required to elucidate whether the modulation of irisin is caused by the inherent mechanisms of underlying diseases, e.g., genetic or inflammatory causes, or whether muscle and bone damage per se influence the circulating levels of this myokine. These studies would help to establish a scientific consensus with regard to the impact of irisin on bone homeostasis in healthy individuals and a possible therapeutic option to treat bone disease.

## Figures and Tables

**Figure 1 jcm-11-06863-f001:**
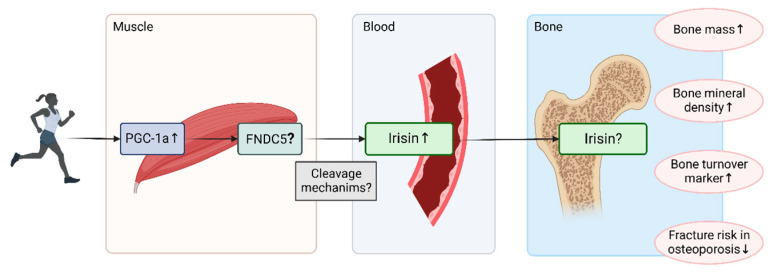
**Irisin and bone in humans.** Exercise increases the expression of peroxisome proliferator-activated receptor gamma coactivator 1-alpha (PGC-1a), a transcriptional regulator of fibronectin type III domain-containing protein 5 (FNDC5), in muscle tissue, as well as circulating levels of the myokine irisin. An upregulation of FNDC5 has not yet been confirmed in the muscle tissue of humans, but only in rodent models, and the exact cleavage mechanisms of FNDC5 into irisin, as well as the local concentrations or secretion of irisin in bone tissues, need further exploration. High serum irisin levels positively correlate with bone mass, bone mineral density, and bone turnover in healthy children and adults, while they are negatively associated with osteoporotic fracture risk. Created with Biorender.com (accessed on 2 March 2022).

## Data Availability

Data were retrieved through a search of electronic databases (PubMed/MEDLINE).

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
