# Peer review of "Irisin and Bone in Sickness and in Health: A Narrative Review of the Literature"

_jcm, 2022, doi:10.3390/jcm11226863_

Round 1
Reviewer 1 Report
In this review, the authors propose a well-structured overview of studies in human subjects that have shown a reduction in serum irisin levels with aging and in the context of bone disease, in both primary and secondary osteoporosis. The authors appropriately point out that it should be kept in mind that previous evidence had questioned whether the detection of this myokine by ELISA, Western blot, and mass spectrometry could be compromised by methodological problems not providing consistently reproducible data.
The only less clear point in this review is included in lines 51-66 in which the authors should provide a plausible explanation of the results obtained on mouse models in which a difference in bone phenotype was observed between exogenous treatment with irisin and overexpression/knocking out of FNDC5.
Author Response
We thank the reviewer for the positive evaluation of our manuscript and for this comment.
As stated in line 64-67: “Also in vitro studies show controversial findings [17,21] implying that the transmembrane receptor FNDC5 itself and its cleavage product, the circulating myokine irisin, might act differentially and independently from each other depending on the targeted cell type and tissue context.”
We connected the paragraphs and rephrased the beginning of the sentence to draw a clearer connection between in vivo results and the explanation. “In line with in vivo studies, in vitro experiments show also controversial outcomes [17,21] implying that the transmembrane receptor FNDC5 itself and its cleavage product, the circulating myokine irisin, might act differentially and independently from each other depending on the targeted cell type and tissue context.”
Reviewer 2 Report
Minor comments
- Line 28. Osteoporosis was defined by the NIH Consensus Development Panel on Osteoporosis Prevention D Osteoporosis prevention, diagnosis, and Therapy (JAMA. 2001;285:785–95).
- Recently published review on the role of irisin in physiology and pathology should be cited (Liu S, Cui F, Ning K, Wang Z, Fu P, Wang D, et al. Role of Irisin in Physiology and Pathology. Frontiers in endocrinology (2022) 13. doi: 10.3389/fendo.2022.962968).
Author Response
We thank the reviewer for the clarifying comments and have modified the References accordingly.